# Chordal-GCN: Exploiting Sparsity in Training Large-Scale Graph Convolutional Networks

## Abstract

Despite the impressive success of graph convolutional networks (GCNs) on numerous applications, training on large-scale sparse networks remains challenging. Current algorithms require large memory space for storing GCN outputs as well as all the intermediate embeddings. Besides, most of these algorithms involves either random sampling or an approximation of the adjacency matrix, which might unfortunately lose important structure information. In this paper, we propose Chordal-GCN for semi-supervised node classification. The proposed model utilizes the *exact* graph structure (*i.e.*, without sampling or approximation), while requires limited memory resources compared with the original GCN. Moreover, it leverages the sparsity pattern as well as the clustering structure of the graph. The proposed model first decomposes a large-scale sparse network into several small dense subgraphs (called *cliques*), and constructs a clique tree. By traversing the tree, GCN training is performed clique by clique, and connections between cliques are exploited via the tree hierachy. Furthermore, we implement Chordal-GCN on large-scale datasets and demonstrate superior performance.

## 1 Introduction

Graph convolutional network (GCN) (Kipf & Welling, 2017) is a generalization of convolutional neural networks (CNNs) (LeCun & Bengio, 1998) to the graph structure. For a given node, the graph convolution operation aggregates the embeddings (features) of its neighbors, followed by a non-linear transformation. By stacking multiple graph convolutional layers, one can learn node representations by utilizing features of its distant neighborhood. The original GCN model, as well as its numerous variations, has shown great success in a variety of applications, including semi-supervised node classification (Kipf & Welling, 2017), inductive node embedding (Hamilton et al., 2017), link prediction (van den Berg et al., 2017), and knowledge graphs (Schlichtkrull et al., 2018).

Despite the success of GCNs, training GCNs on large-scale graphs remains challenging due to the memory issue: we need to store all the parameters and outputs of GCN. Thus, the memory space scales linearly in the size of graph while quadratically in the feature dimension (Chiang et al., 2019; Zou et al., 2019). This prevents applications of GCN on many real-world networks, where the graphs usually contain millions or even billions of nodes.

Methods aimed at large-scale training have been proposed and can be roughly divided into two categories: (1) sampling-based methods and (2) clustering-based methods. For sampling-based methods, only a few neighbors for every node will be sampled in every GCN layer, and thus the size of intermediate embeddings for every layer will be reduced for each mini-batch. Works in this track include Hamilton et al. (2017); Chen et al. (2018a;b); Zou et al. (2019). However, ignorance of some neighbors might lose important structure information, which is the main drawback of all the sampling methods. Another direction of research notices the sparsity of real-world networks and exploits the clustering structure of the graph. For example, Cluster-GCN (Chiang et al., 2019) separates the graph into several clusters, and in every iteration of training, only one or a few clusters are picked to calculate the stochastic gradient for the mini-batch. However, Cluster-GCN ignores all the inter-cluster links, which are not negligible in many real-world networks. For example, Figure 1 shows the sparsity pattern of three citation networks. We first rearrange the vertices via an approximate minimum degree (AMD) ordering algorithm (Amestoy et al., 1996), and then observe a nice arrow pattern in the adjacency matrices. This indicates the existence of some *highly-cited papers*

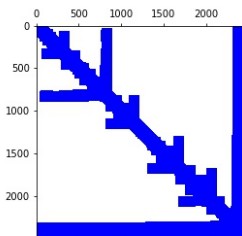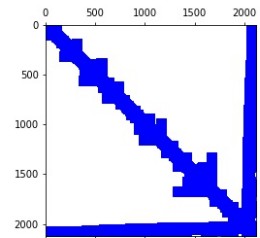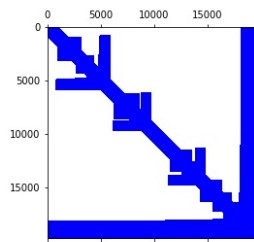

Figure 1: Adjacency matrices for the largest connected component (LCC) of cora (left), citeseer (middle), and pubmed (right). The LCCs contain the majority of the graph, so deleting other nodes does not affect the overall arrow pattern. For simplicity, we treat them as undirected graphs here. Similar patterns are also observed in many other real-world networks (Dong et al., 2019).

(in the right bottom corner), which have impacts on multiple communities. In Cluster-GCN, these highly-cited papers are randomly put into one community and the adjacency matrix is approximated by a block-diagonal matrix; *i.e.*, Cluster-GCN removes the fletching part of the arrow patterns in Figure 1. This approximation ruins the beautiful arrow pattern, and thus ignores the multi-community influence of some seminal papers.

The above difficulties can be easily tackled by the *Chordal-GCN*, a novel clustering-based method for the semi-supervised node classification task. Recall that Cluster-GCN ignores all the inter-cluster links and trains each cluster separately; comparatively, in Chordal-GCN, we keep all the links, train every cluster separately, and at the same time capture the connections between clusters by an additional loss term. This *partially separable* training process can be achieved with the help of chordal sparsity theory (Vandenberghe & Andersen, 2015): we first decompose a large-scale sparse graph into several small dense subgraphs, and we construct a tree of which the nodes are the subgraphs. Note that two subgraphs are adjacent in the tree if they share some vertices. In the training process, we traverse the tree from leaf to root; and when training on a certain subgraph, we minimize the usual GCN loss, plus an additional term called *consistency loss*. With the consistency loss, messages in the children subgraphs can be passed to their parent, and thus the relationship between subgraphs is leveraged via the hierarchy of the tree. Therefore, Chordal-GCN exploits the sparsity pattern as well as the clustering structure of the graph without any approximation or random sampling, while requires similar memory space to Cluster-GCN.

Our contribution is summarized as follows:

- We propose Chordal-GCN for semi-supervised node classification on large-scale sparse networks. The proposed model fully exploits the *exact* graph structure, while requires limited memory usage on large-scale graphs (much smaller than the original GCN).
- Chordal-GCN is able to train a large-scale graph in a partially separable manner; *i.e.*, in every iteration, the training is performed on a subgraph, and the connections between subgraphs are handled by a consistency loss.
- We analyze the memory and time complexity of Chordal-GCN and compare them with other state-of-the-art GCN models. Also, we evaluate the performance of Chordal-GCN on benchmark datasets and demonstrate superior performance in large-scale datasets.

## 2 BACKGROUND AND RELATED WORK

### 2.1 SEMI-SUPERVISED NODE CLASSIFICATION WITH GCNS

Suppose $G = (V, E)$ is an undirected graph with $|V| = n$ and the $n \times n$ symmetric matrix $A$ is the adjacency matrix. Every node $k$ is associated with a feature vector $x_k \in \mathbb{R}^d$, and all the feature vectors are stored in the rows of $X \in \mathbb{R}^{n \times d}$. An $L$-layer graph convolutional network (GCN) (Kipf & Welling, 2017) has the layer-wise propagation rule:

$$H^{(l+1)} = \sigma_l(\bar{A}H^{(l)}W^{(l)}),$$

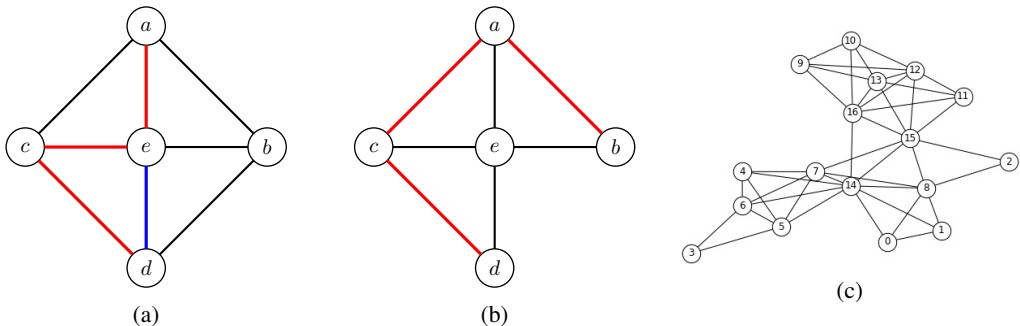

Figure 2: (a) An example of non-chordal graph. The edge $\{d, e\}$ is a chord in the path $(a, e, c, d)$. This graph is non-chordal because the cycle $(a, b, d, c, a)$ is chordless. (b) An example of chordal graph. The path $(b, a, c, d)$ is chordless. (c) A chordal graph with 17 vertices.

where $W^{(l)} \in \mathbb{R}^{d_l \times d_{l+1}}$ is a weight matrix. The matrix $\bar{A} = \tilde{D}^{-1/2} \tilde{A} \tilde{D}^{-1/2}$ is the normalized adjacency matrix, where $\tilde{A} = I + A$ and $\tilde{D} \in \mathbb{R}^{n \times d}$ is diagonal with $\tilde{D}_{ii} = \sum_j \tilde{A}_{ij}$. The matrix $H^{(l)} \in \mathbb{R}^{n \times d_l}$ are the activations in the $l$-th layer ($H^{(0)} = X$ and $d_0 = d$). The function $\sigma_l$ is the activation function for layer $l$. The output of an $L$-layer GCN is an $n \times p$ matrix $H^{(L)}$, of which the $k$-th row is the predicted label for node $k$. Later in the paper, we will train GCN using only a subgraph $G(\gamma)$, so the notation $y_k^{\langle \text{pred} \rangle}(W; X_{\gamma,:}, \bar{A}_{\gamma\gamma})$ is used to indicate the predicted label of node $k$, using the coefficient matrices $W = (W^{(0)}, \ldots, W^{(L-1)})$, the submatrix $\bar{A}_{\gamma\gamma}$, and the corresponding feature vectors $X_{\gamma,:}$.

Semi-supervised node classification is a popular application of GCN. When applying GCN for this task, we minimize the loss function

$$\mathcal{L}^{\langle \text{gcn} \rangle}(W; X, \bar{A}) = \sum_{k \in V^1} \ell_1\big(y_k^{\langle \text{true} \rangle}, y_k^{\langle \text{pred} \rangle}(W; X, \bar{A})\big), \tag{1}$$

to learn the weight matrices $W = (W^{(0)}, \ldots, W^{(L-1)})$. In the formula, $y_k^{\langle \text{true} \rangle}$ is the given true label for node $k$, and $V^1 \subseteq V$ is the subset of vertices with given true labels. The function $\ell_1(\cdot, \cdot)$ is a loss function, and usually it is the cross entropy loss.

Much work has been done to apply GCN on large-scale datasets. For the sampling-based methods, different rules for random selection have been proposed: uniform sampling as in Hamilton et al. (2017), or importance sampling in Chen et al. (2018a). On the other hand, clustering-based methods become more interesting in the recent literature. Besides Cluster-GCN (Chiang et al., 2019), Graph Partition Neural Network (GPNN) Liao et al. (2018) also partitions the entire graph into small clusters, and the GCN model alternatively propogates within-cluster and between-cluster. In addition, clustering structure can also be utilized implicitly as a low-rank approximation of the graph Laplacian. This idea is exploited in LanczosNet (Liao et al., 2019) as well as the Lovász convolutional network (Yadav et al., 2019).

## 2.2 PRELIMINARIES ON CHORDAL GRAPHS

Chordal sparsity has been a classical topic in graph theory and found useful applications in various fields, including database theory (Beeri et al., 1983), probabilistic networks (Pearl, 1988; Cowell et al., 1999; Darwiche, 2009), linear algebra (Rose, 1970), combinatorial optimization (Golumbic, 2004; Gavril, 1972), semidefinite optimization (Andersen et al., 2010; 2013). In this paper, we will use chordal sparsity theory to exploit the clustering structure. So now we show that a chordal graph can be separated into cliques, and that these cliques form a clique tree with desirable properties.

**Ordered undirected graph.** Given an undirected graph $G = (V, E)$ with $|V| = n$, we can order the vertices from 1 to $n$, and then we refer to a vertex in $V$ by its order $i \in [n] = \{1, 2, \ldots, n\}$. In addition, an index set $\gamma \subset [n]$ represents a subset of vertices in $V$.

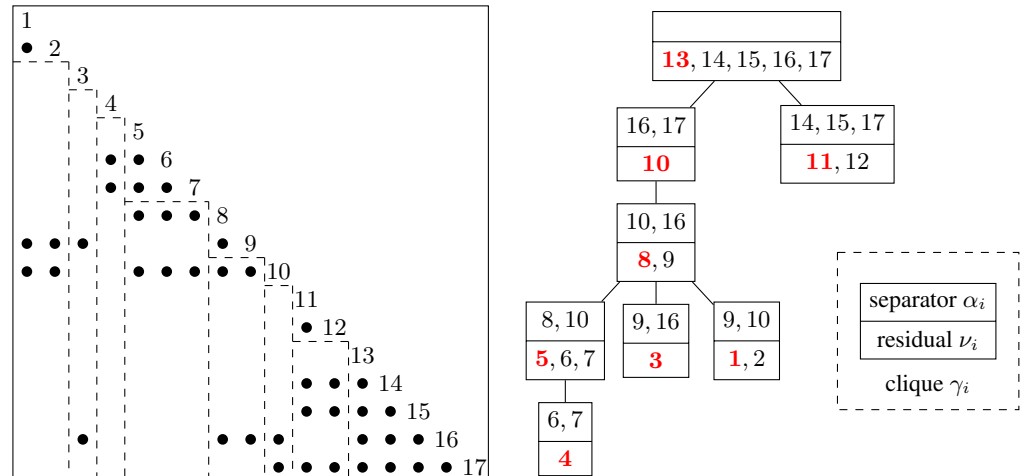

Figure 3: *Left.* The (symmetric) adjacency matrix of the graph $G$ shown in Figure 2c. (For simplicity we only show the lower triangular part.) A bullet in the $(i, j)$th entry means nodes $i$ and $j$ are adjacent. The dashed lines separate the supernodes (clique residuals) in the clique tree. *Right.* The corresponding postordered clique tree $T^c$. Every node in $T^c$ is a clique in $G$, and the red index $i$ in every clique indicates the clique representative. The dashed box shows an example clique.

**Cliques.** A *complete graph* is a graph in which every pair of distinct vertices is adjacent. A *clique* $\gamma \subseteq V$ of a graph $G$ is a subset of vertices that induces a maximal complete subgraph. The graph induced by $\gamma$ is denoted $G(\gamma) = (\gamma, E(\gamma))$ and has edge set $E(\gamma) = \{\{v, w\} \in E \mid v, w \in \gamma\}$.

**Chordal graph.** A *path* between $v_0$ and $v_k$ is a sequence of distinct vertices $(v_0, v_1, v_2, \ldots, v_k)$ with $\{v_i, v_{i+1}\} \in E$ for $i = 0, 1, \ldots, k - 1$. For example, the graph in Figure 2a has a path $(a, e, c, d)$. A *chord* in a path of an undirected graph is an edge between non-consecutive vertices on the path. For example, in Figure 2a, the edge $\{d, e\}$ is a chord in the path $(a, e, c, d)$, but in Figure 2b, the path $(b, a, c, d)$ is chordless. An undirected graph is *chordal* if every cycle of length greater than three has a chord. Figure 2b and 2c show two examples of chordal graphs, while the graph in Figure 2a is non-choral.

**Postordered clique tree.** Given a (connected) chordal graph $G = (V, E)$, we can always construct a *clique tree* $T^c$: the nodes of $T^c$ are the cliques $\gamma \subseteq V$ and two cliques are adjacent in $T^c$ if they share some vertices in $V$. In addition, we can arbitrarily pick a clique as the root of the tree, and then every non-rooted clique can be partitioned into *clique separators* and *clique residuals*. The clique separator is the intersection of the clique with its parent clique in $T^c$. The clique residual is also called the *(maximal) supernode*, and by definition, all the supernodes form a partition of $V$.

More interestingly, we can always find an ordering of vertices $V$ so that the clique tree $T^c$ satisfies the following properties.

1. The higher neighborhood of every vertex is complete: $j, k \in \mathrm{adj}^+(i)$ implies $\{j, k\} \in E$, where the higher neighborhood of vertex $i$ is defined as $\mathrm{adj}^+(i) = \{j \mid \{i, j\} \in E, \ j > i\}$. This means that every clique and supernode in $T^c$ can be written as $\gamma_i$ and $\nu_i$, respectively, where $i \in V$ has the lowest order in the clique (and supernode). We denote by $i$ the *representative* of the supernode $\nu_i$, by $\alpha_i = \gamma_i \backslash \nu_i$ the corresponding clique separator, and by $V^c$ the set of all representatives. The index set $\alpha_i$ is empty if $i$ is the root of $T^c$.

2. The elements of each supernode are numbered consecutively: $\nu_i = \{i, i + 1, \ldots, i + n_i\}$ with $n_i = |\nu_i - 1|$.

3. All the representatives $i \in V^c$ are topologically ordered: $i < p(i)$ if $i \in V^c$ and $p(i)$ is the representative of the parent of $\gamma_i$ in $T^c$.

An ordering satisfying the above three properties is called *topological postordering*, and a clique tree $T^c$ with certain topological postordering is referred to as a *postordered clique tree*. Figure 3

shows a supernode partition of the chordal graph in Figure 2c with a topological postordering. (This example is adapted from the survey paper Vandenberghe & Andersen (2015).) In this example,

$$\gamma_5 = \{5, 6, 7, 8, 10\}, \qquad \nu_5 = \{5, 6, 7\}, \qquad \alpha_5 = \{8, 10\}.$$

Given a chordal sparsity graph $G = (V, E)$, there exist efficient algorithms (in $O(|E|)$) to find a topological postordering, and to generate a postordered clique tree $T^c$ together with the parent function $p(i)$ and all the index sets (*i.e.*, $\gamma_i$, $\nu_i$, and $\alpha_i$ for all $i \in V^c$) (Lewis et al., 1989; Pothen & Sun, 1990). The whole process is called *chordal decomposition* (Vandenberghe & Andersen, 2015).

## 3 CHORDAL-GCN

Now we are ready to utilize the chordal decomposition technique in training large-scale sparse networks. We note that most real-world networks are indeed sparse, but not chordal. So we first introduce some data preprocessing steps, followed by the detailed description of Chordal-GCN. We also analyze the time and memory complexity of our model, and compare it with other GCN models.

### 3.1 PREPROCESSING

**Reordering and chordal extension.** Chordal extension describes the idea of transforming a non-chordal graph $G = (V, E)$ into a chordal graph $G' = (V, E')$ by adding edges, *i.e.*, $E \subseteq E'$. Unfortunately, to find the minimum fill-in is claimed to be an NP-complete problem (Rose et al., 1976; Yannakakis, 1981), and is not always desirable due to algorithm complexity. However, given an ordering of vertices, it is practical to find a *minimal* chordal extension (Ohtsuki, 1976; Rose, 1974). (It is called minimal if the removal of any added edge results in a non-chordal graph.) That is to say, every minimal chordal extension is associated with certain vertex ordering. Thus, the reordering of vertices is important, especially when we prefer a sparse extended graph. So in this work, we use an approximate minimum degree (AMD) ordering algorithm (Amestoy et al., 1996) to first *reorder* the vertices, and then find a minimal chordal extension.

**Clique merging.** Often, the resulting extension contains many small cliques and it would be more efficient to merge some neighboring cliques, as reported in Ashcraft & Grimes (1989). In this work, traversing the tree from leaf to root, we greedily merge clique $\gamma_i$ with its parent $\gamma_{p(i)}$ if $|\nu_i| \leq \tau$ and $|\nu_{p(i)}| \leq \tau$ where $\tau$ is a pre-defined threshold. We denote the resulting graph as $G'' = (V, E'')$.

**Chordal decomposition.** Given the graph $G''$, we can now construct the postordered clique tree $T^c$ (together with all the index sets ($\gamma_i$, $\nu_i$, and $\alpha_i$), and the parent function $p(i)$). With a little abuse of notation, we also use $T^c$ to indicate the postordered clique tree of $G$. In other words, the preprocessing steps *reorder* and *partition* the vertex set $V$, which helps us identify the clique $\gamma_i$ and the corresponding induced subgraph $G(\gamma_i)$. Although chordal extension and clique merging add edges into the graph, it will not affect the training process of GCN because we will use the subgraph $G(\gamma_i)$, not $G''(\gamma_i)$.

Lastly, we emphasize that, although $G(\gamma_i)$ no longer represents a complete subgraph, we still refer to $\gamma_i$ as *clique* to distinguish from *clusters* or *communities*: in the chordal decomposition, one node $k \in V$ is allowed to exist in multiple cliques while in community detection or graph clustering, one node appears in only one cluster.

### 3.2 CHORDAL-GCN: META ALGORITHM

With the postordered clique tree $T^c$ in hand, we are able to train a large-scale sparse graph in a *partially separable* manner, *i.e.*, clique by clique. Suppose we traverse the tree $T^c$ in the topological order, *i.e.*, from leaf to root. At clique $\gamma_i$, the loss function consists of two parts. The first part is the usual GCN loss defined in (1), but using only the induced subgraph $G(\gamma_i)$ instead of the entire $G$:

$$\mathcal{L}^{\langle \text{gcn} \rangle}(\cdot; X_{\gamma_i,:}, \bar{A}_{\gamma_i \gamma_i}) = \sum_{k \in V^l \cap \gamma_i} \ell_1 \big( y_k^{\langle \text{true} \rangle}, y_k^{\langle \text{pred} \rangle}(\cdot; X_{\gamma_i,:}, \bar{A}_{\gamma_i \gamma_i}) \big),$$

where $X_{\gamma_i,:}$ takes the rows of $X$ indexed by $\gamma_i$ and $\bar{A}_{\gamma_i \gamma_i}$ is the submatrix of $\bar{A}$ with rows and columns indexed by $\gamma_i$.

---

**Algorithm 1** Chordal-GCN.

---

1: **Input.** A normalized adjacency matrix $\bar{A}$, a postordered clique tree $T^{\mathrm{c}}$, feature matrix $X$, and true labels $y_k^{\langle\mathrm{pred}\rangle}$ for $k \in V^1$.
2: **Output.** The coefficient matrices $W = (W^{(0)}, \ldots, W^{(L-1)})$.
3: Randomly initialize $W^{[0]}$.
4: **for** $t = 0, 1, 2, \ldots$ **do**
5:     **for** $i \in V^{\mathrm{c}}$ in topological order **do**
6:         Compute the loss by (4), *i.e.*,

$$
\begin{aligned}
\mathcal{L}_i(W) = &\sum_{k \in V^1 \cap \gamma_i} \ell_1\big(y_k, y_k^{\langle\mathrm{pred}\rangle}(W; X_{\gamma_i,:}, \bar{A}_{\gamma_i\gamma_i})\big) \\
&+ \sum_{j \in \mathrm{ch}(i)} \sum_{k \in \alpha_j} \ell_2\big(y_k^{\langle\mathrm{pred}\rangle}(W; X_{\gamma_i,:}, \bar{A}_{\gamma_i\gamma_i}), y_k^{\langle\mathrm{pred}\rangle}(W^{[t]}; X_{\gamma_j,:}, \bar{A}_{\gamma_j\gamma_j})\big).
\end{aligned}
\tag{2}
$$

7:         Perform a gradient descent step $W^{[t+1]} = W^{[t]} - \eta \nabla \mathcal{L}_i(W^{[t]})$.
8:     **end for**
9: **end for**

---

The second part of the loss is the *consistency loss*. When we arrive at clique $\gamma_i$, we have traversed all its children in the tree $T^{\mathrm{c}}$. This means that some nodes in $\gamma_i$ have already been visited; namely, those clique separators of $\gamma_i$'s children in $T^{\mathrm{c}}$. For clarity, we denote by $\mathrm{ch}(i)$ the set of children representatives of $\gamma_i$; *i.e.*, $j \in \mathrm{ch}(i)$ if $j$ is a representative (*i.e.*, $j \in V^{\mathrm{c}}$) and $\gamma_j$ is a child of $\gamma_i$ in $T^{\mathrm{c}}$. Then all the nodes in $\alpha_j$ for $j \in \mathrm{ch}(i)$ have already been visited when we arrive at $\gamma_i$. Thereby, for $k \in \alpha_j$ and $j \in \mathrm{ch}(i)$, we hope that the predicted label $y_k^{\langle\mathrm{pred}\rangle}(\cdot; X_{\gamma_i,:}, \bar{A}_{\gamma_i\gamma_i})$ using the current clique $\gamma_i$ is consistent with the previous prediction $y_k^{\langle\mathrm{pred}\rangle}(\tilde{W}; X_{\gamma_j}, \bar{A}_{\gamma_j\gamma_j})$ using clique $\gamma_j$ and the most recent parameter $\tilde{W}$. Hence, the consistency loss for training clique $\gamma_i$ can be formulated as

$$
\mathcal{L}^{\langle\mathrm{cons}\rangle}(\cdot; X_{\gamma_i,:}, \bar{A}_{\gamma_i\gamma_i}) = \sum_{j \in \mathrm{ch}(i)} \sum_{k \in \alpha_j} \ell_2\big(y_k^{\langle\mathrm{pred}\rangle}(\cdot; X_{\gamma_i,:}, \bar{A}_{\gamma_i\gamma_i}), y_k^{\langle\mathrm{pred}\rangle}(\tilde{W}; X_{\gamma_j,:}, \bar{A}_{\gamma_j\gamma_j})\big), \quad (3)
$$

where $\ell_2(\cdot, \cdot)$ is a loss function, for example, the KL divergence. Note that $\tilde{W}$ is not treated as the variable of $\mathcal{L}^{\langle\mathrm{cons}\rangle}$; *i.e.*, the term $y_k^{\langle\mathrm{pred}\rangle}(\tilde{W}; X_{\gamma_j,:}, \bar{A}_{\gamma_j\gamma_j})$ is considered constant when we take the gradient of $\mathcal{L}^{\langle\mathrm{cons}\rangle}$.

Therefore, when training the clique $\gamma_i$, we compute the total loss

$$
\mathcal{L}_i(\cdot) = \mathcal{L}^{\langle\mathrm{gcn}\rangle}(\cdot; X_{\gamma_i,:}, \bar{A}_{\gamma_i\gamma_i}) + \mathcal{L}^{\langle\mathrm{cons}\rangle}(\cdot; X_{\gamma_i,:}, \bar{A}_{\gamma_i\gamma_i}), \tag{4}
$$

and perform a gradient descent step $W^+ = W - \eta \nabla \mathcal{L}_i(W)$ where $\eta$ is the learning rate. The whole algorithm is summarized in Algorithm 1. Note that the gradient descent step in line 7 of Algorithm 1 can be replaced with any other accelerated version, for example, Adam (Kingma & Ba, 2015).

### 3.2.1 CONNECTION TO THE ORIGINAL GCN

We note that Algorithm 1 is essentially the mini-batch gradient descent method applied to the loss

$$
\mathcal{L}^{\langle\mathrm{cgcn}\rangle}(W) = \sum_{i \in V^{\mathrm{c}}} \mathcal{L}_i(W) \tag{5}
$$

with $\gamma_i$ taken as the mini-batch. Clearly, it is equal to the original GCN loss (1) if we have only one clique (*i.e.*, $V^{\mathrm{c}} = \{1\}$ and thus $\gamma_1 = V$). Also, when $\bar{A}$ is block diagonal, Chordal-GCN is equivalent to Cluster-GCN because the consistent loss (3) vanishes.

However, in the general case, without any assumption on the graph structure or the GCN structure, it is impossible to show the equivalence between Chordal-GCN and GCN. From another view of point, the non-convex functions (1) and (5) don't share any global minimizer. For example, consider the chordal graph with two overlapping dense principal submatrices. We can easily choose the set of true labels $y^{\langle\mathrm{true}\rangle}$ such that (1) and (5) don't have the same minimizer.

Table 1: Time and memory complexity of GCN training algorithms. $L$ is the number of layers, $n$ is the number of nodes, $\|A\|_0$ is the number of nonzeros in the adjacency matrix $A$, and $d$ is the number of features. For simplicity we assume the number of features is fixed for all layers, *i.e.*, $d_l = d$ for $l = 1, 2, \ldots, L$. For SGD-based methods, $b$ is the batch size and $r$ is the number of sampled neighbors per node. Also, $c_1$ is the maximum cluster size in Cluster-GCN while $c_2$ is the maximum clique size in our method. For memory complexity, $Ld^2$ is used to store the parameters $\{W^{(l)}\}_{l=0}^{L-1}$ and the other term is for storing embeddings $\{H^{(l)}\}_{l=1}^{L}$. For simplicity we omit the memory for storing the graph or subgraphs since they are fixed and usually not the main bottleneck.

| Complexity | GCN (Kipf & Welling, 2017) | GraphSAGE (Hamilton et al., 2017) | FastGCN (Chen et al., 2018a) | VR-GCN (Chen et al., 2018b) | Cluster-GCN (Chiang et al., 2019) | Chordal-GCN |
|---|---|---|---|---|---|---|
| Time | $O(L\|A\|_0d + Lnd^2)$ | $O(r^L nd^2)$ | $O(Lrnd^2)$ | $O(L\|A\|_0d + (L+r^L)nd^2)$ | $O(L\|A\|_0d + Lnd^2)$ | $O(L\|A\|_0d + Lnd^2)$ |
| Memory | $O(Lnd + Ld^2)$ | $O(br^Ld + Ld^2)$ | $O(Lbrd + Ld^2)$ | $O(Lnd + Ld^2)$ | $O(Lc_1d + Ld^2)$ | $O(Lc_2d + Ld^2)$ |

### 3.2.2 TIME AND MEMORY EFFICIENCY

Following Chiang et al. (2019), we report the time and memory complexity in Table 1. Although our method has the same time complexity as Cluster-GCN in big-O notation, Chordal-GCN needs an extra forward propagation on the children cliques in order to compute the predicted labels $y_k^{\langle \text{pred} \rangle}(W^{[t]}; X_{\gamma_j,:}, \bar{A}_{\gamma_j \gamma_j})$ in (2). The extra forward propagation step would be the only price paid for single node being in multiple subgraphs. As we will see in the numerical experiments, this additional cost is minimal and worthwhile, especially in citation networks with an arrow pattern.

In addition, the memory bottleneck for Chordal-GCN and Cluster-GCN depend on the maximum cluster size and maximum clique size, respectively. Since these two numbers depend totally on the hyperparameters, *i.e.*, number of clusters in METIS algorithm and the merging threshold $\tau$, it is difficult to compare analytically. But both are much smaller than the memory usage of the original GCN, which is desirable.

## 4 EXPERIMENTS

### 4.1 EXPERIMENT SETTINGS

We evaluate our proposed model for the semi-supervised node classification task on four public datasets: cora, citeseer, pubmed, and reddit. We compare Chordal-GCN with the following state-of-the-art GCN training algorithms for comparison: GCN (Kipf & Welling, 2017), GraphSAGE (Hamilton et al., 2017), FastGCN (both uniform and importance sampling) (Chen et al., 2018a), VR-GCN (Chen et al., 2018b), and Cluster-GCN (Chiang et al., 2019). For all the methods we use the same GCN structure (*i.e.*, $L = 2$), the same label rate, and Adam optimizer (Kingma & Ba, 2015) with learning rate 0.001, dropout rate as 20%, weight decay as zero. For the baselines, we use the implementations provided by the authors, and follow the default parameter settings in these models. Other implementation details are included in appendix A.

For all the methods and datasets, we conduct training for 10 times and take the mean of the evaluation results. Also, we stop training when the validation accuracy does not increase a threshold (0.01) for 10 epoches, and choose the model with the highest validation accuracy as convergent point. We use the following metrics to evaluate the performance of all methods.

- **Accuracy:** The micro F1 score of the test data at the convergent point.
- **Memory usage:** The maximum GPU memory occupied by tensors.
- **Epoch time and number of epoches:** Time to run an epoch and total number of epoches before convergence. Note that in Chordal-GCN, an epoch is a traverse of the clique tree, rather than one clique.

### 4.2 NUMERICAL RESULTS

We summarize all the results in Table 2. In the three citation networks (cora, citeseer, and pubmed), Chordal-GCN achieves the highest accuracy among all baseline models. This desirable result is due to the exploitation of the exact sparsity pattern and clustering structure.

Table 2: Comparsion of Chordal-GCN with original GCN, GraphSAGE, FastGCN with uniform/importance sampling, VR-GCN, and Cluster-GCN. We report the micro F1 score (%), memory usage (MB), time per epoch (ms), and number of epoches. Only GCN and Chordal-GCN utilize the exact graph, while other baseline methods lose some graph structure information.

| Dataset | Method | F1 (%) | Mem. (MB) | Epoch time (ms) | # epoches |
|---|---|---|---|---|---|
| Cora (2708) | GraphSAGE | 78.4 | 565.12 | 44.26 | 11 |
| | FastGCN (uniform) | 78.2 | 28.42 | 27.10 | 88 |
| | FastGCN (importance) | 83.5 | 28.42 | 30.41 | 84 |
| | VR-GCN | 81.6 | 4.60 | 14.90 | 64 |
| | Cluster-GCN | 68.2 | 36.14 | 62.65 | 8 |
| | GCN | 78.4 | 20.23 | 5.35 | 119 |
| | Chordal-GCN | **81.2** | 5.69 | 90.90 | 49 |
| Citeseer (3327) | GraphSAGE | 61.8 | 932.14 | 37.97 | 60 |
| | FastGCN (uniform) | 70.2 | 98.93 | 111.00 | 53 |
| | FastGCN (importance) | 72.2 | 98.93 | 104.90 | 61 |
| | VR-GCN | 71.2 | 7.01 | 32.00 | 35 |
| | Cluster-GCN | 62.8 | 60.35 | 82.09 | 5 |
| | GCN | 70.5 | 52.16 | 6.42 | 95 |
| | Chordal-GCN | **74.1** | 8.47 | 38.91 | 57 |
| Pubmed (19717) | GraphSAGE | 76.8 | 475.84 | 43.57 | 47 |
| | FastGCN (uniform) | 77.4 | 107.20 | 25.41 | 52 |
| | FastGCN (importance) | 78.0 | 107.20 | 22.00 | 47 |
| | VR-GCN | 78.4 | 14.49 | 25.21 | 33 |
| | Cluster-GCN | 71.2 | 25.35 | 59.84 | 10 |
| | GCN | 78.0 | 66.03 | 7.52 | 109 |
| | Chordal-GCN | **80.2** | 15.37 | 55.94 | 83 |
| Reddit (232965) | GraphSAGE | 93.1 | 1192.00 | 185.22 | 62 |
| | FastGCN (uniform) | 92.9 | 1656.14 | 15.83 | 64 |
| | FastGCN (importance) | 93.0 | 1722.05 | 21.60 | 31 |
| | VR-GCN | 93.2 | 1054.21 | 5041.32 | 32 |
| | Cluster-GCN | **96.1** | 231.32 | 2407.61 | 43 |
| | GCN | 93.9 | 2992.12 | 261.81 | 190 |
| | Chordal-GCN | 94.2 | 2501.23 | 987.12 | 410 |

In terms of memory space, Chordal-GCN is superior to the original GCN, just as expected. All the sampling-based methods require limited memory space because they reduce the number of intermediate node embeddings. More interestingly, Chordal-GCN uses less memory space than Cluster-GCN (except in reddit). This is because the Cluster-GCN implementation uses a few clusters in one epoch, and thus the memory bottleneck depends on the sum of the selected clusters, rather than the maximum cluster. The abnormal memory cost of Chordal-GCN in reddit dataset is attributed to our clique merging heuristic. The maximal clique in reddit has size $220, 069$, almost $95\%$ of the vertices. Splitting the graph into more cliques will help reduce the memory requirement, while increases the epoch time: we train smaller but more cliques in one epoch.

Lastly, we emphasize that the comparison of epoch time can never be fair: one epoch in Chordal-GCN means one traverse of the clique tree while in other methods one epoch is one mini-batch. This explains the long epoch time of Chordal-GCN.

## 5 CONCLUSION

We propose Chordal-GCN for semi-supervised node classification on large-scale sparse networks. The proposed model exploits the sparsity pattern and clustering structure of the graph, and utilizes the *exact* graph structure (*i.e.*, without sampling or approximation). Moreover, the memory usage

of Chordal-GCN is limited because the training is performed in a partially separable manner; *i.e.*, clique by clique. Experiment results demonstrate that Chordal-GCN achieves the best test accuracy with much smaller memory cost on benchmark datasets.

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

## A    IMPLEMENTATION DETAILS

**Hardware and software.**    We use Tesla V100 GPU, Intel Xeon CPU E5-2698 (2.20GHz) with 500GB. We use PyTorch 1.2.0 in our implmentation and Tensorflow 1.13 to test the baselines.

**Datasets.**    The cora, citeseer, and pubmed datasets are from https://github.com/tkipf/gcn. The reddit data is from https://snap.stanford.edu/graphsage. We report the data statistics in Table 3.

Table 3: Data statistics

| Dataset | Nodes | Edges | Classes | Features | Label rate |
|---|---|---|---|---|---|
| Cora | 2,708 | 5,429 | 7 | 1,433 | 140 (5.1%) |
| Citeseer | 3,327 | 4,732 | 6 | 3,703 | 120 (3.6%) |
| Pubmed | 19,717 | 44,338 | 3 | 500 | 60 (0.3%) |
| Reddit | 232,965 | 11,606,919 | 41 | 602 | 152,410 (65.4%) |

Table 4: Parameters in Chordal-GCN

| Dataset | Threshold for merging ($\tau$) | Number of cliques | Maximum clique size | Minimum clique size |
|---|---|---|---|---|
| cora | 100 | 19 | 446 | 125 |
| citeseer | 200 | 6 | 512 | 227 |
| pubmed | 2,000 | 8 | 6,105 | 2,132 |
| reddit | 10,000 | 126 | 220,069 | 1,001 |

**Preprocessing.** In Chordal-GCN, we use the AMD ordering (Amestoy et al., 1996) implemented in the python package CVXOPT (Andersen et al., 2015). The minimal chordal extension and the chordal decomposition are performed using the python package chompack (Andersen & Vandenberghe, 2015). We also use the clique merging heuristic described in section 3.1. The threshold $\tau$ is different in different datasets, and mainly depends on the size of the graph. We report the parameters in Table 4.

**GCN structure.** We implement our model with PyTorch. We use a two-layer GCN with fixed parameter size 128.

To calculate the meory usage, we use `tf.contrib.memory_BytesInUse()` for TensorFlow and `torch.cuda.memory_allocated()` for PyTorch.

# B ADDITIONAL EXPERIMENTS

## B.1 ABLATION STUDY

We evaluate the contribution of the consistency loss $\mathcal{L}^{\langle \text{cons} \rangle}$ (3) here. We train Chordal-GCN with and without $\mathcal{L}^{\langle \text{cons} \rangle}$ on the cora, citeseer, and pubmed datasets, and plot the validation accuracy versus the number of epoches. Figure 4 shows that the consistency loss $\mathcal{L}^{\langle \text{cons} \rangle}$ correctly captures the connection between cliques, and that improve the performance of Chordal-GCN. This explains why our model performs consistently better than Cluster-GCN: Cluster-GCN ignores those inter-cluster links and thus loses the connections between clusters. In the pubmed dataset, Chordal-GCN without consistency loss is even overfitting: the validation accuracy decreases after several epoches.

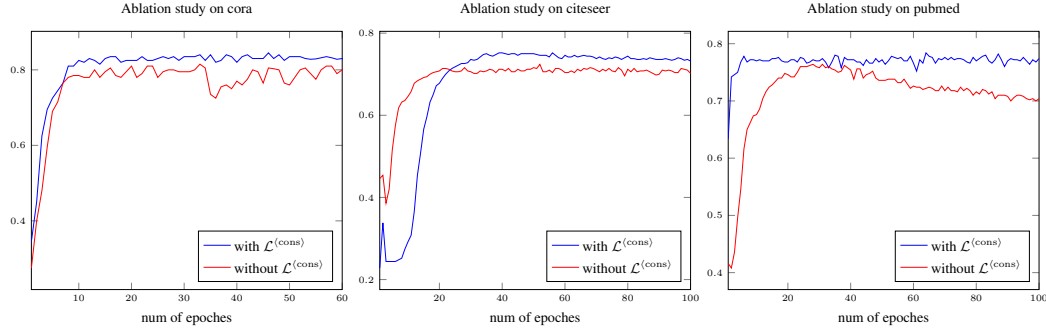

Figure 4: Validation accuracy of Chordal-GCN on the datasets: cora, citeseer, and pubmed. The blue curve shows the validation accuracy with the consistent loss $\mathcal{L}^{\langle \text{cons} \rangle}$ while the red curve shows that without $\mathcal{L}^{\langle \text{cons} \rangle}$.

## B.2 TRAINING DEEP GCNS

In this section, we study the performance of deeper Chordal-GCN, and explore how time and memory cost scales with GCN depth. As shown in Figure 5, both memory and time cost scale linearly with the number of layers. This result is consistent with theoretical analysis, and shows that Chordal-GCN is applicable to training deeper GCNs.

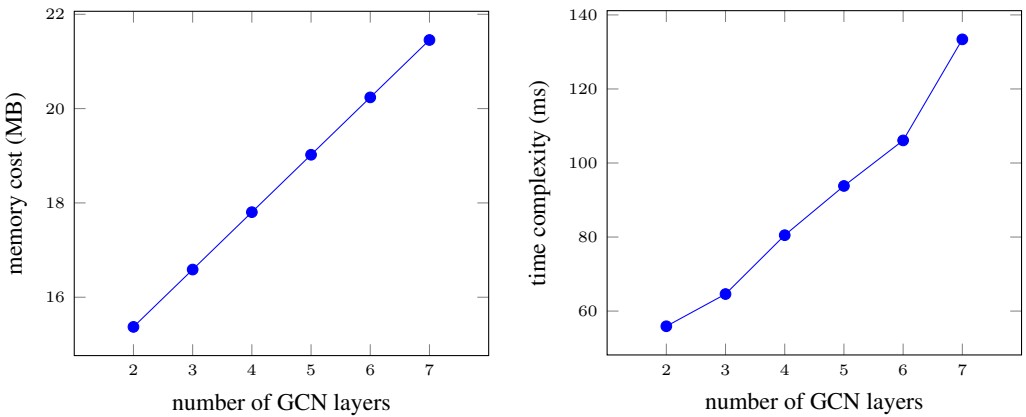

Figure 5: Memory (*left*) and time (*right*) versus number of GCN layers. Both curves scale linearly, which is promising and consistent with theoretical analysis.

