# OpenReview forum: "Chordal-GCN: Exploiting sparsity in training large-scale graph convolutional networks"
_ICLR.cc/2020/Conference — Reject_

### Official Review · AnonReviewer2 · 2019-10-11
**Official Blind Review #2**

**Rating:** 1

**Review:**

In this paper, the authors propose a new method referred to Chordal-GCN to optimize memory usage in large graphs. The authors borrow the ideas from Chordal Sparsity theory to first build a client tree. Then mini-batch updates are carried out individually on each clique from the leaves following the GCN loss. The authors add an additional consistency loss between shared node with children cliques. Experiments are carried out in four networks with comparison to several baselines.

Strength:
1. The authors study an interesting and important problem to reduce memory usage for GCN in large-scale graphs. The usage of chordal sparsity is interesting and innovative.
2. The authors carry out ablation study on the consistency loss components in the algorithm.

Weakness:
1. One major concern is that the reduction in memory usage is not large enough to justify the huge increment in running-time. For example, on Cora dataset, the memory is reduced by 4x while the running time is 16x compared to vanilla GCN.
2. It is not clear why Chordal-GCN can achieve better accuracy compared to vanilla GCN. Since Chordal-GCN serves as approximation for GCN as the authors claim that using entire graph will provide better accuracy. As a result, the vanilla GCN is expected to achieve better accuracy. It would be better if the authors could provide more intuition and explanations.
3. The evaluation does not take the graph preprocessing into consideration. The authors should report the time and memory taken to carry out the preprocessing steps as well.
4. For most real-world large-scale industry networks, it is hard to fit the graph into memory. Though the GCN training part could run in distributed way, it is not clear how to efficiently build the clique tree in similar method.
5. Given the main purpose of the algorithm is to reduce memory usage for large-scale networks, it is expected to see experiments on larger graphs where the large memory footprint becomes a real issue.

Detailed comments:
1. The description in Section 2.2 is not very clear. It would be better if the authors could provide a more detailed introduction to clique tree and the algorithms used.
2. For the Chordal-GCN in algorithm 1, for epoch 2 onwards, do we also add consistent-loss when training leaves as well?


**Experience Assessment:**

I have published one or two papers in this area.

**Review Assessment: Checking Correctness Of Derivations And Theory:**

I assessed the sensibility of the derivations and theory.

**Review Assessment: Checking Correctness Of Experiments:**

I carefully checked the experiments.

**Review Assessment: Thoroughness In Paper Reading:**

I read the paper at least twice and used my best judgement in assessing the paper.

---

> ### Author Response · Authors · 2019-11-15
> **Thank you for your comments.**
>
> We thank the reviewer for the constructive reviews. We addressed the questions and concerns of the reviewer accordingly in the following.
>
> Weakness:
> 1. One major concern is that the reduction in memory usage is not large enough to justify the huge increment in running-time. For example, on Cora dataset, the memory is reduced by 4x while the running time is 16x compared to vanilla GCN.
>
> Among all the baselines, vanilla GCN always takes the minimum time. Chordal-GCN is slower because Chordal-GCN executes more than one GCN model in every epoch, i.e., one GCN for one clique in Chordal-GCN. However, the running time of Chordal-GCN is comparable to other baselines.
>
> Actually, in Chordal-GCN there is a trade-off between the memory and the training time. If we divide the graph into more cliques, then the required memory, which is determined by the largest clique size, will be reduced. But at the same time, the training time will increase because every clique corresponds to one GCN model. We will explore this balance issue in the future.
>
> 2. It is not clear why Chordal-GCN can achieve better accuracy compared to vanilla GCN. Since Chordal-GCN serves as approximation for GCN as the authors claim that using entire graph will provide better accuracy. As a result, the vanilla GCN is expected to achieve better accuracy. It would be better if the authors could provide more intuition and explanations.
>
> Both Chordal-GCN and vanilla GCN use the entire graph---without sampling or any other approximation, so we don’t treat our model as an ‘‘approximation’’ of GCN. Instead, Chordal-GCN modifies the GCN training procedure such that the training can be performed in a distributed manner.
>
> An intuition is provided in paragraph 3 of the introduction section: in most citation networks, highly-cited papers should have impacts on multiple communities. Thus, Chordal-GCN treats the ‘‘inter-cluster’’ links and the ‘‘intra-cluster’’ links differently (while GCN doesn’t make a distinction). In training a particular node, Chordal-GCN considers the influence of its neighbors in the same clique first, and the impacts from other cliques are handled via the consistency loss. This different treatment might explain an increase in the accuracy.
>
> 3. The evaluation does not take the graph preprocessing into consideration. The authors should report the time and memory taken to carry out the preprocessing steps as well.
>
> The most expensive step in preprocessing is building the clique tree, of which the time and memory are both linear in the number of nonzeros in the adjacency matrix A.
>
> 4. For most real-world large-scale industry networks, it is hard to fit the graph into memory. Though the GCN training part could run in distributed way, it is not clear how to efficiently build the clique tree in similar method.
> The SOTA methods for building clique trees [1,2] can be easily extended to a distributed version. Basically, one can start with the node with the largest ID, and then find the clique this node is in. Other neighbors of this node (with largest ID) are certainly in the children clique. Thereby, every time we only need to know the neighbors of the current node, instead of the entire graph.
>
> 5. Given the main purpose of the algorithm is to reduce memory usage for large-scale networks, it is expected to see experiments on larger graphs where the large memory footprint becomes a real issue.
>
> Thank you for your suggestion. we plan to include this result in our final draft.
>
> Detailed comments:
>
> 1. The description in Section 2.2 is not very clear. It would be better if the authors could provide a more detailed introduction to clique tree and the algorithms used.
>
> Thank you for your advice and sorry for the confusion. The algorithm for building the clique tree is too sophisticated to explain in two pages, and is not the focus of our paper. So we choose not to include in the preliminaries. We hope the unclearness in Section 2.2 does not affect the understanding of our main model.
>
> 2. For the Chordal-GCN in algorithm 1, for epoch 2 onwards, do we also add consistent-loss when training leaves as well?
>
> No. The consistency loss involves the current clique and its children in the clique tree. Since leaves of a tree never have a child (ch(i) is a empty set in Eq. (2)), there is no consistency loss when training leaves.
>
> [1] P. Buneman. A characterization of rigid circuit graphs. Discrete Mathematics. 9:205-212, 1974.
> [2] F. Gavril. The intersection graphs of subtrees in trees are exactly chordal graphs. Journal of Combinatorial Theory Series B, 16:47-56, 1974.

---

### Official Review · AnonReviewer1 · 2019-10-23
**Official Blind Review #1**

**Rating:** 6

**Review:**

This paper leverages the clique tree decomposition of the graph and design a new variant of GCN which does graph convolution on each clique and penalize the inconsistent prediction made on separators of each node and its children. Experiments on citation networks and the reddit network show that the proposed method is efficient.

Overall, this paper could be a significant contribution on improving GCN, with the caveat for some clarifications on the model and experiments. Given these clarifications in an author response, I would be willing to increase the score.

Pros:

1, I like the idea of exploiting graph decomposition. In my opinion, it may not only improve the scalability but also help the model learn representations which better capture the structure or speed up the learning process. It would be great if authors could show some evidence along this line.

2, The examples in Figure 2 and 3 are very helpful in understanding the concepts related to the clique tree decomposition.

3, The summarization of time and memory complexity is very helpful in comparing different models.

4, I read the detailed questions and responses in the open review. It helps me understand more details about the experiments. Besides the typo of Table 2, I tend to believe that the experimental setup is reasonable and results are convincing although I did not run the code by myself.

Cons & Questions:

1, The main motivation of exploiting the graph decomposition is to save memory such that GCN could be applied to large scale graphs without sacrificing the structural information. However, the scale of the largest experiments is still less impressive. To strengthen the paper, it would be great to try larger graph datasets which have been used in the literature.

2, I am confused by the writing on the final prediction made by the model. In particular, do you only keep the prediction of residual or do you average the predictions on the separators? It may be interesting to explore different ways of making predictions based on this decomposition based inference. In general, it would be great to separate the writing of loss (learning) and prediction (inference).

3, Why does Chordal-GCN take significant more epochs than GCN on Reddit and less epochs on all other datasets?

Suggestion:

1, I think the clique tree is very similar if not the same with the junction tree given the node ordering (see section 2.5.2 of [1]). It would be great to discuss the relationship between your chordal graph representation and the tree decomposition used by the probabilistic inference algorithms of graphical models. From the perspective of complexity, the junction tree method and yours both highly depend on the tree-width. Also, linking to probabilistic inference could help better motivate the method since tree-based inference algorithm is shown to converge faster in the literature.

2, It would be great to discuss and or compare with [2] as it uses graph partition algorithms to get clusters and apply GNN with a propagation schedule which alternates between within-cluster and between-cluster. It is closely related to the chordal-GCN as it uses the decomposition of graph clustering directly rather than the clique tree. Decomposition like multiple overlapping spanning trees are also studied in [2].

[1] Wainwright, M.J. and Jordan, M.I., 2008. Graphical models, exponential families, and variational inference. Foundations and Trends® in Machine Learning, 1(1–2), pp.1-305.

[2] Liao, R., Brockschmidt, M., Tarlow, D., Gaunt, A.L., Urtasun, R. and Zemel, R., 2018. Graph partition neural networks for semi-supervised classification. arXiv preprint arXiv:1803.06272.

======================================================================================================

Thank authors for the thorough reply! After I read authors' rebuttal and other reviewers' comments, I would like to keep my original rating. Again, I like this idea and believe better exploiting structure in the propagation could improve the inference in many ways. I hope authors could keep improving it, e.g., better motivating the proposed method (memory saving is just one angle which sometimes may need more engineering work to fully verify) and change the experiments accordingly.

**Experience Assessment:**

I have published in this field for several years.

**Review Assessment: Checking Correctness Of Derivations And Theory:**

I carefully checked the derivations and theory.

**Review Assessment: Checking Correctness Of Experiments:**

I assessed the sensibility of the experiments.

**Review Assessment: Thoroughness In Paper Reading:**

I read the paper at least twice and used my best judgement in assessing the paper.

---

> ### Author Response · Authors · 2019-11-15
> **Responds to Review #1**
>
> We thank the reviewer for the constructive reviews. We addressed the questions and concerns of the reviewer accordingly in the following.
>
> Weakness
> 1, Thank you for your suggestion. We plan to include this result in our final draft.
>
> 2, In our current submission, the testing procedure is identical to the vanilla GCN because storing the entire graph is not the main problem (storing GCN parameters and embeddings needs much more space than storing the graph). That is why we only include the training phase in Algorithm 1. Nevertheless, your suggestion might motivate further study in another direction: Comparison between the labels predicted by the entire graph and those predicted by every clique might enlighten better understanding on the graph structure. For example, when the predictions are different, does it mean that the current clique has a ‘‘negative’’ impact on predicting this node; and is this negative impact due to noise in the network, or in the feature matrix? We believe all these questions are interesting but challenging, and thus beyond the scope of this paper.
>
> 3, The outlier in the Reddit dataset is attributed to a bad choice of hyperparameters. With our current hyperparameters, Reddit is decomposed into a giant clique and some small cliques. This unbalanced decomposition cause the abnormal results. We will improve the results in the Reddit dataset by finding better hyperparameters in our final draft.
>
> Suggestion:
> 1, The clique tree in our paper is indeed the same as the junction tree in [1] because any chordal graph must have a clique tree with the running intersection property (in Definition 2.1 of [1]). The only minor difference is that, in our definition, every node in the clique tree is a ‘‘maximal’’ clique while in some literature, the junction tree does not require maximality. In our humble opinion, the term ‘‘clique tree’’ is more used by the chordal sparsity community while the term ‘‘junction tree’’ is often used by researchers in graphical models and probabilistic networks.
>
> We would have motivated our idea from the perspective of message passing and probabilistic inference. However, we try to avoid the illusion that Chordal-GCN is a new, sophisticated variant of GCN model; instead, we would like to persuade readers that Chordal-GCN is a modification in the GCN training procedure. After all, the idea of linking GCN to probabilistic inference itself is interesting and worth further exploration.
>
> Moreover, we appreciate it a lot if you can notify us with any efficient C implementation of building junction trees, and the tree-based inference algorithms. We are happy to compare the performance of chordal decomposition and the junction tree decomposition in [1]. Although our model still remains the same, the actual running time (especially the preprocessing time) might be improved dramatically.
>
> 2, Thank you for bringing the paper [2] into our attention. In our biased opinion, it is more similar to Cluster-GCN [3]. GPNN [2] finds a partition of the nodes and treats inter-cluster and intra-cluster links differently while Chordal-GCN somehow finds a ‘‘partition’’ of the edges and treats clique separators and residuals in a different manner. Building minimum spanning trees in [2] is also used for node partition purpose.
>
> After all, it is a very interesting and highly related paper. We have added the citation and will consider it as an important baseline in our final draft.
>
> Last but not least, we would like to thank you again for your affirmation in our paper. We also appreciate your helpful and insightful comments and suggestions.
>
> [1] Wainwright, M.J. and Jordan, M.I., 2008. Graphical models, exponential families, and variational inference. Foundations and Trends® in Machine Learning, 1(1–2), pp.1-305.
> [2] Liao, R., Brockschmidt, M., Tarlow, D., Gaunt, A.L., Urtasun, R. and Zemel, R., 2018. Graph partition neural networks for semi-supervised classification. arXiv preprint arXiv:1803.06272.
> [3] W.-L. Chiang, X. Liu, S. Si, Y. Li, S. Bengio, and C.-J. Hsieh. Cluster-GCN: an efficient algorithm for training deep and large graph convolutional networks. KDD 2019.

---

### Official Review · AnonReviewer3 · 2019-10-23
**Official Blind Review #3**

**Rating:** 3

**Review:**

The authors propose Chordal-GCN which is based on the chordal decomposition method post-ordered clique tree and propagates the features based on the order within each subgraph in order to reduce memory usage. The authors show that Chordal-GCN outperforms GCN [1] on all four datasets and argue that Chordal-GCN reduces memory usage.
The idea of using Chordal graphs to GCN is novel and interesting. However, my main concern lies in the experiment results.

1) To my best knowledge, the proposed Chordal- match SOTA results on Cora, Citeceer, and Pubmed. However, since these datasets are small and easy to run, I would like to see the mean and standard deviation of the accuracy of all models you ran. Can you also provide the results of the commonly used "random split setting"[1]?

2) What is the epoch time of the Chordal-GCN? Can you also report it in Table 2? Without including the pre-processing time, we don't know the overall training time of the method.

3) Given that the main concern is the memory usage, the authors should compare to a strong baseline, SGC [2], which is a linear classifier trained on top of propagated features with memory/space complexity O(d) when using mini-batch training. This is much smaller than the proposed method O(Lc_2d + Ld^2).
Also, SGC is at least two magnitudes faster to train (2.7s vs 0.987*410=367.8s + unknown pre-processing time) and more accurate (94.9 vs 94.2) than the proposed Chordal-GCN on the largest Reddit dataset. The authors emphasize that the proposed method is scalable. Please compare it to SGC in Table 2.
Nevertheless, there is some chance that the authors can apply the same method to SGC and speed it up further as long as the preprocessing time is relatively small.

4) Based on Table 2, Cluster-GCN uses less memory and is more accurate and faster to train than Chordal-GCN. Can you justify why people should use the proposed method instead?

5) There are some missing citations. These papers [3,4,5,6] achieved previous SOTA results and should be included in the Tables.

References:
[1] Kipf and Welling: Semi-Supervised Classification with Graph Convolutional Networks (ICLR 2017)
[2] Wu et al.: Simplifying Graph Convolutional Networks (ICML 2019)
[3] Klicpera et al.: Predict then Propagate: Graph Neural Networks meet Personalized PageRank (ICLR 2019)
[4] Gao and Ji: Graph U-Nets (ICML 2019)
[5] Zhang et al.: GaAN: Gated Attention Networks for Learning on Large and Spatiotemporal Graphs (UAI 2018)
[6] Fey: Just Jump: Dynamic Neighborhood Aggregation in Graph Neural Networks (ICLR-W 2019)

**Experience Assessment:**

I have published one or two papers in this area.

**Review Assessment: Checking Correctness Of Derivations And Theory:**

I assessed the sensibility of the derivations and theory.

**Review Assessment: Checking Correctness Of Experiments:**

I carefully checked the experiments.

**Review Assessment: Thoroughness In Paper Reading:**

I read the paper at least twice and used my best judgement in assessing the paper.

---

> ### Author Response · Authors · 2019-11-15
> **Thank you for your comments**
>
> We thank the reviewer for the constructive reviews. We addressed the questions and concerns of the reviewer accordingly in the following.
>
> 1) To my best knowledge, the proposed Chordal- match SOTA results on Cora, Citeceer, and Pubmed. However, since these datasets are small and easy to run, I would like to see the mean and standard deviation of the accuracy of all models you ran. Can you also provide the results of the commonly used "random split setting"[1]?
>
> Ans: The split in [1] is chosen by the authors, and thus fixed. Existing work [7] has proven that this split of dataset has a significant influence on the classification result. We follow the random held-out strategy and randomly split the dataset for training and test multiple times. This random split strategy is used in all baseline models, and we think this is a fair and consistent setting.
>
> 2) What is the epoch time of the Chordal-GCN? Can you also report it in Table 2? Without including the pre-processing time, we don't know the overall training time of the method.
>
> Ans: We have already reported the epoch time in Table 2.
>
> For the preprocessing, the main bottleneck is to build the clique tree, of which the time complexity is linear in the number of nonzeros in the adjacency matrix. In our current implementation, the preprocessing mainly depends on the python package Chompack, and thus it takes more time than Cluster-GCN---as the preprocess in Cluster-GCN depends on a C package.
>
> 3) Given that the main concern is the memory usage, the authors should compare to a strong baseline, SGC [2], which is a linear classifier trained on top of propagated features with memory/space complexity O(d) when using mini-batch training. This is much smaller than the proposed method O(Lc_2d + Ld^2).
> Also, SGC is at least two magnitudes faster to train (2.7s vs 0.987*410=367.8s + unknown pre-processing time) and more accurate (94.9 vs 94.2) than the proposed Chordal-GCN on the largest Reddit dataset. The authors emphasize that the proposed method is scalable. Please compare it to SGC in Table 2.
> Nevertheless, there is some chance that the authors can apply the same method to SGC and speed it up further as long as the preprocessing time is relatively small.
>
> Ans: Thank you for your suggestion. In our biased opinion, Chordal-GCN, as well as all the baselines we used, is a modification in the training phase of the vanilla GCN. Comparatively, SGC should be treated as a totally different graph neural network model. In light of the model structure, SGC only has one layer, and the graph structure $A$ is aggregated with the feature information $X$ before training. This preprocessing step facilitates training but is ‘‘expensive’’ in nature: In preprocessing, one should compute $A^K X$ (in our notation). Computing matrix power $A^K$ involves eigenvalue decomposition of $A$, which is $O(n^3)$. The short preprocessing time in the SGC paper is attributed to highly optimized numerical linear algebra package BLAS and LAPACK, which are written in Fortran language. If clique tree building can be carried out in Fortran or C, the preprocessing time will be limited ($O(||A||_0)$) in Chorda-GCN.
>
> 4) Based on Table 2, Cluster-GCN uses less memory and is more accurate and faster to train than Chordal-GCN. Can you justify why people should use the proposed method instead?
>
> Ans: In the first three datasets (Cora, Citeseer, Pubmed), Chordal-GCN uses less memory and achieves better results than Cluster-GCN. We also provide an intuitive explanation in Section 1.
>
> The outlier in Reddit it due to a wrong selection of hyperparameters. With our current hyperparameters, Reddit is decomposed into a giant clique and some small cliques. This unbalanced decomposition causes the abnormal memory cost. We will improve the results in the Reddit dataset by finding better hyperparameters in our final draft.
>
> 5) There are some missing citations. These papers [3,4,5,6] achieved previous SOTA results and should be included in the Tables.
>
> Ans: Thank you for bringing these papers into our attention. However, there are so many papers on GCN models and we can only compare the most related ones. We argue that Chordal-GCN, as well as all the baselines we use, focuses on the training phase of the vanilla GCN. Thereby, variants of GCN models are not considered as baselines in our submission, despite their interestingness.
>
> References:
> [1-6] same as above
> [7] Oleksandr Shchur, Maximilian Mumme, Aleksandar Bojchevski, Stephan Günneman: Pitfalls of Graph Neural Network Evaluation (NeurIPS workshop 2018)

---

### Public Comment · ~Qiang_Liu7 · 2019-10-13
**Experimental results seem wrong in table 2.**

I have carefully reviewed the code provided, and found something might be wrong:

(1)	In the load_data function in utils.py, the authors randomly select 300 nodes for training, 200 nodes for validation and 1000 nodes for testing. This is totally different from the settings in table 3, which is the same as in the original papers of the baselines. The author claims that “For the baselines, I use the implementations provided by the authors, and follow the default parameter settings in these models” in section 4.1. So, for example, according to table 3, 140 constant labeled nodes in the Cora dataset are used to train ordinary GCN. However, according to the code provided by the authors, 300 randomly labeled nodes are used to train Chordal-GCN, which means the authors used more supervised information. So, I don’t think results in table 2 can give a fair comparison.
So, what is the performance of Chordal-GCN under a fair experimental setting?
I revised the code of the authors, and randomly selected only 140 labeled nodes for training of Chordal-GCN. I ran the code 10 times, and the averaged accuracy is 76.9, which is much smaller than that reported in the paper. Then, I randomly selected 300 labeled nodes for training of the original GCN, and the averaged accuracy in 10 times is 84.6, which is much larger than that reported in the paper. These results are totally different from the results in table 2.
(2)	According to the code, in each time of training, the training, validation and testing nodes are randomly selected. So, testing samples are different in each time of running the code. Tough the authors claim “we conduct training for 10 times and take the mean of the evaluation results”, I don’t think the results are meaningful.
(3)	The metric used in the paper is f1-score, while the evaluation used in the code is accuracy. This is confusing.
(4)	In the load_data function, the authors claim “cora only for now”. The authors gave a new data format for the Cora dataset, but there lacks the corresponding new format for the other three datasets. And the load_data function in the code can only read the new data format. Considering the original Cora, Citeseer and Pubmed datsets share the same data format and are easy to use, I wonder why the authors presented a new format. And if the new format of other datasets is not ready, how was the experiments conducted?
So, I wrote a new load_data function for reading the original format of Cora, Citeseer and Pubmed. When conducting experiments on Cora, I can obtain the same results with the two load_data functions. However, when conducting experiments on Pubmed, the accuracy is only ~0.5.
(5)	The reported results of Cluster-GCN are much lowers than those in the original paper.
(6)	According to the code, during the testing procedure, the whole adjacency matrix is still used for the inference of final prediction. This doesn’t match the propose of the paper when conducting large-scale online prediction. And if we perform inference on each batch, I believe the performance of Chordal-GCN will be even l`ower.

In summary, I think the experimental results in this paper are hardly convincing.

---

> ### Author Response · Authors · 2019-10-17
> **Thank you for your interest and comments.**
>
> Thank you for your interest and comments. We answer your questions one by one below.
>
> (1) Ans: The above link is the code with hyperparameters tuned for Cora dataset, and it follows Kipf's Pytorch implementation [1]. In order to test on other datasets, please follow the hyperparameter setup reported in Table 4.
>
> For Cora dataset, our provided code and the result in Table 2 do use 300 training nodes, not 140. The number 140 in the last column of Table 3 is a typo. We appreciate it a lot for your careful proofreading. We have re-run our code on Cora using the common label rate. The following result still shows the superior performance of Chordal-GCN compared with GCN, which is consistent with the results in our paper.
>
>  	Label rate   Chordal-GCN   GCN
> Cora	140          81.20 	78.36
> Cora	300          85.51 	82.22
>
> We have carefully checked other results: we follow the experimental setting for other datasets in Table 3 and we can reproduce the results in Table 2.
>
> (2) Ans: Even though using fixed data split is a convention, we think the trained model will be biased to the current test set. So we follow the random held-out strategy and randomly split the dataset for training and test multiple times. This random split strategy is used in all baseline models, and we think this is a fair and consistent setting.
>
> (3) Ans: Accuracy is exactly the same as micro-F1 score in any single-label classification task, with the only exception that some predicted labels are out of range (i.e., the class is an integer between 1 and 10 but one prediction is 11). This can be easily shown as follows: Assume there are n nodes and k classes, and then
>
> precision = (TP_1+...+TP_k)/(TP_1+...+TP_k+FP_1+...+FP_k) = (TP_1+...+TP_k)/N = accuracy
> recall	= (TP_1+...+TP_k)/(TP_1+...+TP_k+FN_1+...+FN_k) = (TP_1+...+TP_k)/N = accuracy
> F1    	= (2 x precision x recall) / (precision + recall) = accuracy.
>
> Since all the datasets used in our paper are single-labeled, we follow the convention to report the micro-F1 score.
>
> (4) Ans: The data format used in the uploaded code follows Kipf's implementation [1], and it is widely used in GCN related research.
>
> For other datasets, please follow the hyperparameters reported in Table 4: the merging thresholds are different for different datasets.
>
> (5) Ans: The reported result for Reddit is consistent with Cluster-GCN [6]. The authors' code [3] does not provide the implementation for other datasets. So we write our own data loader for Cora, Citeseer, and Pubmed datasets. The GCN structure is different from [6], and we don’t know the details about other hyperparameters used for these three datasets. These two reasons might explain the inconsistency of the results in our paper and [6].
>
> (6) Ans: The whole adjacency matrix is indeed used in the testing procedure, and this is the convention used in the original GCN [1,2,4] as well as its variants [5,6].
>
> Original GCN can provide prediction for unseen nodes. The prediction will be accurate only if the network does not change too much; otherwise, everything has to be retrained. More importantly, such prediction still needs to use the entire graph to compute the latent representation for the new node and thus the predicted label.
>
> Although Chordal-GCN is proposed for large-scale training, it can also be used for online prediction. Since online inference occurs in the testing procedure and does not affect the training process, it won’t be the memory bottleneck for Chordal-GCN.
>
> [1] https://github.com/tkipf/pygcn
> [2] https://github.com/tkipf/gcn
> [3] https://github.com/google-research/google-research/tree/master/cluster_gcn
> [4] T. N. Kipf, M. Welling, Semi-Supervised Classification with Graph Convolutional Networks (ICLR 2017)
> [5] J. Chen, J. Zhu, L. Son, Stochastic Training of Graph Convolutional Networks with Variance Reduction (ICML 2018)
> [6] W.-L. Chiang et al., Cluster-GCN: An Efficient Algorithm for Training Deep and Large Graph Convolutional Networks (KDD 2019)
> [7] J. Chen, T. Ma, C. Xiao, FastGCN: Fast Learning with Graph Convolutional Networks via Importance Sampling (ICLR 2018)

---

### Decision · Program_Chairs · 2019-12-19

**Decision:**

Reject

**Comment:**

The submission is proposed a rejection based on majority review.